# Tracking striking algal changes over the last ~400 years using subfossil pigments in a high mountain lake (Sierra Nevada, Spain): Have we entered an unprecedented era?

Joana Llodrà-Llabrés<sup>1,#</sup>, Carmen Pérez-Martínez<sup>1,#</sup>, Teresa Vegas-Vilarrúbia<sup>2</sup>, John P. Smol<sup>3</sup>, Carsten Meyer-Jacob<sup>4</sup>, Javier Sigro<sup>5</sup>, Teresa Buchaca<sup>6</sup>

Correspondence to: Carmen Pérez-Martínez (cperezm@ugr.es)

Abstract. Remote aquatic ecosystems have been shown to be affected by the rapid intensification of human-driven climate change, along with increasing atmospheric nutrient deposition. There is an increasing body of evidence from paleolimnology that indicates changes in the composition of diatoms due to both factors. However, there is a paucity of studies that examine changes in the composition of the overall algal community over extended periods of time. This study investigates shifts in pigment assemblage composition and algal biomass over approximately the past 430 years, using high-resolution, well-dated sediment cores from Borreguil Lake, a high-altitude lake in the Sierra Nevada Mountains (Southern Spain). Significant changes in both algal biomass and community composition were observed throughout the core, with notable intensification since the ca. 1970s. These changes appear to be a regional response primarily driven by climate and atmospheric aerosols. Algal biomass exhibited two significant peaks approximately between 1740-1840 and from 1970 to the present, with the latter period reaching unprecedented concentrations. Algal composition exhibited two major shifts: one around 1840 and another in the 1970s. From the bottom to the top, these shifts were characterized by an increase in cyanobacteria (indicated by aphanizophyll and scytonemin), cryptophytes (indicated by alloxanthin), and green algae (indicated by lutein and zeaxanthin), at the expense of diatoms (indicated by diatoxanthin). Statistical analyses revealed that both algal biomass and composition were strongly influenced by warming temperatures, reduced precipitation, and enhanced Saharan dust deposition. In particular, the increase in nitrogen-fixing cyanobacteria (indicated by aphanizophyll) since the 1970s has led to previously unrecorded nitrogen fixation in the lake. This is probably due to reduced nitrogen availability linked to enhanced Saharan phosphorus inputs. The observed changes in the algal community, including the significant increase of cyanobacteria biomass, are unprecedented in the last ~400 years in the Sierra Nevada lakes and are likely occurring in other

<sup>&</sup>lt;sup>1</sup>Department of Ecology and Institute for Water Research, University of Granada, 18071 Granada, Spain

<sup>&</sup>lt;sup>2</sup>Department of Evolutionary Biology, Ecology and Environmental Sciences, Universitat de Barcelona, Av. Diagonal 643, 08028 Barcelona, Spain

<sup>&</sup>lt;sup>3</sup>Paleoecological Environmental Assessment and Research Laboratory, Department of Biology, Queen's University, Kingston, Ontario K7L 3N6, Canada

<sup>&</sup>lt;sup>4</sup>IRF, Université du Québec en Abitibi-Témiscamingue, Rouyn-Noranda, Canada

<sup>&</sup>lt;sup>5</sup>Centre for Climatic Change (C3), Research Institute in Sustainability, Climate Change and Energy Transition, University Rovira i Virgili, C. Joanot Martorell 15, 43480 Vilaseca, Tarragona, Spain

<sup>&</sup>lt;sup>6</sup>Integrative Freshwater Ecology (CEAB-CSIC), Blanes, Spain

<sup>15 \*</sup>These authors contributed equally to this work

Mediterranean lake regions, particularly in oligotrophic lakes. Projected increases in global temperatures and Saharan dust deposition will likely continue to affect the ecological condition of these ecosystems.

### 1 Introduction

ecosystem dynamics.

High mountain lakes are especially vulnerable to climate warming (Moser et al., 2019; Sorvari and Korhola, 1998). Their physical constraints, nutrient limitations, and low productivity and biodiversity amplify the effects of environmental changes. Furthermore, stressors linked to climate warming, such as temperature, are intensified in these ecosystems (Pepin et al., 2015). As a result, these remote lakes are considered biospheric sentinels of global change (Adrian et al., 2009). In addition to changes caused by climate warming, remote lakes are sometimes exposed to increased atmospheric nitrogen deposition due to human activities (Bergström and Jansson, 2006). Furthermore, recent data indicate that phosphorus deposition in alpine ecosystems is increasing (Brahney et al., 2014; Camarero and Catalán, 2012; Jiménez et al., 2018), mainly due to windblown dust from distant phosphorus sources.

Of particular concern is the case of high mountain lakes in the Mediterranean region, as they face two distinct forms of stress: their high-altitude location and their position within the Mediterranean zone (Nogués-Bravo et al., 2008). Projections for the Mediterranean region have shown projected temperature increases and precipitation decreases (IPCC, 2022; López-Merino et al., 2011). Climate models predict regional warming at rates 20% higher than the global average, along with a

12% reduction in precipitation for a 3°C global temperature rise. As a result, the region is considered a critical 'hotspot' in future climate projections (Giorgi, 2006; IPCC, 2022). Additionally, significant amounts of Saharan dust are transported to the Mediterranean (Salvador et al., 2022).

The Sierra Nevada, the highest range in the Iberian Peninsula and southern Europe, provides a unique setting for studying global change in the Mediterranean basin. With elevations over 3000 m a.s.l., it lies between European and African biogeographic regions, about 60 km from the coast, and experiences a semi-arid Mediterranean climate (Zamora and Oliva, 2022). Due to its proximity to Africa, Sierra Nevada is an ideal reference for studying Saharan inputs, as southeastern Iberia is frequently exposed to Saharan dust intrusions (Morales-Baquero and Pérez-Martínez, 2016; Pulido-Villena et al., 2006). Since 1864 CE, the Sierra Nevada region has warmed by 1.56 °C, mainly due to increased spring and summer temperatures. Summer precipitation has declined, particularly from 1975 to 2020 CE (Sigro et al., 2024). Additionally, Saharan dust input—rich in phosphorus, calcium, and alkalinity (Rogora et al., 2004)—has significantly increased in recent decades (Moulin and Chiapello, 2006; Mulitza et al., 2010; Prospero and Lamb, 2003), further influencing lake chemistry and

Sierra Nevada contains around 50 small, shallow lakes with remote locations, minimal human impact, low primary production, and low alkalinity. These features make them key reference sites for monitoring climate dynamics, global change, and atmospheric factors (Pérez-Martínez et al., 2022; Zamora and Oliva, 2022). The sediment records in these lakes serve as valuable archives of past environmental and ecological conditions, providing insights into historical limnological

changes and shifts in the surrounding catchment and airshed (Jiménez-Moreno et al., 2022; Pérez-Martínez et al., 2022). These long-term records in these sentinel ecosystems are crucial for understanding potential future changes.

Several paleolimnological studies conducted in Sierra Nevada lakes, focusing on the Anthropocene period (last ~200 years), suggest that rising regional air temperatures, reduced precipitation, and increased P- and Ca-rich Saharan dust deposition are the primary drivers of ecological changes observed in six Sierra Nevada lakes. Notably, these climate changes have increased chlorophyll-a concentrations, with more pronounced effects since the ~1970s yr CE and have influenced diatom (Pérez-Martínez et al., 2020b) and cladoceran assemblages (Jiménez et al., 2018). Additionally, shifts in chironomid assemblages in Río Seco Lake (Sierra Nevada) are also linked to these climatic trends (Jiménez et al., 2019). Moreover, previous limnological studies within the Sierra Nevada region have elucidated the impacts of interannual climatic variations, including temperature, rainfall, and Saharan dust deposition, on biogeochemical processes and lake biota (Morales-Baquero et al., 2006; Pérez-Martínez et al., 2007). All these changes align with patterns observed in mountain lakes worldwide, which have undergone significant changes throughout the twentieth century (Baron et al., 2000; Saros et al., 2011; Williamson et al., 2016).

Despite the well-documented impact of climate change on Sierra Nevada lakes, the long-term response of algal community composition remains unknown. Here, we provide detailed analyses of sedimentary pigments, in addition to other biogeochemical indicators (such as stable isotopes and the elemental composition of organic matter), from the Little Ice Age to the present (last ~430 years). Pigment-based methods allow for the reconstruction of taxonomic group abundances that lack morphological remains (Leavitt and Hodgson, 2001). Accordingly, source-specific sedimentary pigments have been used to identify specific algae types (Zhang et al., 2019), based on the premise that each algal class has a characteristic pigment composition (Mackey et al., 1996). Pigments have been shown to be effective taxonomic markers for studying algal communities in both marine (Riegman and Kraay, 2001) and freshwater ecosystems, including eutrophic (Schlüter et al., 2006) and oligotrophic lakes (Brahney et al., 2015a; Buchaca et al., 2005; Oleksy et al., 2020). Sedimentary pigments in remote lakes have linked increased primary production to a variety of factors, including climate warming (Battarbee et al., 2002; Lami et al., 2010; Michelutti et al., 2005; Michelutti and Smol, 2016) and atmospheric phosphorus input (Brahney et al., 2015a). For example, sedimentary carotenoid and chlorophyll-a analysis in a Chinese alpine lake revealed increased productivity and shifts in algal dominance since the 1960s yr CE, driven by dam construction, higher N and P inputs, and climate changes (Zhang et al., 2019). In Sierra Nevada lakes, Jiménez et al. (2018) observed increased sedimentary chlorophyll-a since the 1970s yr CE, linked to rising temperatures and enhanced phosphorus deposition from Saharan dust.

Based on the information outlined above, we hypothesized that recent climate changes—specifically, rising temperatures and decreasing precipitation in the Sierra Nevada—together with increased Saharan dust deposition, have significantly affected the abundance and composition of primary producers in Borreguil Lake, as recorded in sedimentary pigments. The atmospheric input of Saharan-derived phosphorus (P) and calcium (Ca) is particularly relevant given the naturally low levels of these nutrients in the lake (Jiménez et al., 2018; Pérez-Martínez et al., 2020a). Previous studies have documented marked ecological shifts in the lake since the 1960s–1970s yr CE, including a longer ice-free period, rising water temperatures, and

125

increased alkalinity and nutrient concentrations—particularly phosphorus (Jiménez et al., 2018, 2019; Pérez-Martínez et al., 2020b, 2022)—all of which are likely to have influenced both productivity and community structure. In this context, the main objectives of this study were to reconstruct ecological changes in Borreguil Lake over the past ~430 years by examining (a) algal biomass trends, (b) changes in algal community composition through sedimentary pigment analysis, and (c) variations in key biogeochemical variables. By integrating pigment data, paleoenvironmental proxies, and climate records, the study aims to identify long-term patterns in algal abundance and composition, determine the key environmental drivers behind these trends, and establish which algal groups were most responsive to them. Accordingly, we posed the following research questions: (1) Has algal biomass and/or primary production increased over the past ~430 years? (2) Can changes in productivity and biomass be explained by climatic or other environmental variables? (3) Has the composition of the algal community shifted over time? (4) What were the main environmental factors driving these changes? Overall, this research contributes to a deeper understanding of long-term ecological dynamics in alpine lake algal communities in the context of ongoing climatic and atmospheric changes.

### 2 Methodology

### 2.1 Study area

The Sierra Nevada range (Granada, SE Spain) (36° 55′-37° 15′ N, 2° 31′-3° 40′ W) is the highest mountain range in continental Spain and Europe outside the Alps, with a peak elevation of 3482 m asl. The summits experience a high-elevation Mediterranean climate, characterized by warm, dry, and ice-free conditions from June to October. The meteorological station at 2507 m asl reports an annual mean temperature of 4.4°C and 700 mm of precipitation, 80% of which falls as snow between October and April. The area, designated a national park in 1999, is a biodiversity hotspot, containing over 2100 vascular plant taxa, representing approximately 30% of the Iberian flora, with high endemicity and numerous threatened species (Pérez-Luque et al., 2016).

Sierra Nevada hosts approximately 50 small glacial lakes at elevations of 2800–3100 m asl, formed during the glacial retreat after the last glacial cycle (Castillo Martín, 2009). These lakes are typically oligotrophic or oligo-mesotrophic, characterized by cold, oxygen-rich waters with low alkalinity and mineralization. This study focuses on Borreguil Lake (Fig. 1), an oligo-mesotrophic lake at 2980 m asl (37° 03′ 09.53" N, 3° 17′ 59.03" W) with a maximum depth of 2.8 m and an area of 0.18 ha. The lake freezes annually from October to May, with interannual variability. It is fishless, does not thermally stratify in summer, and has a catchment area partly covered by alpine meadows. The lake basin is composed of siliceous mica schist, devoid of carbonated rocks, with permanent inlets and outlets supplying water during the ice-free period. Physicochemical data are available in Jiménez et al. (2018) and Pérez-Martínez et al. (2020b).

Figure 1: A) Location of the study area including the location of Sierra Nevada in Spain (© Google Earth 2019). B) Borreguil Lake (© Google Earth 2019). C) Bathymetric map of the lake (extracted from digitized map of bathymetry report from Egmasa S.A.) The orange dot indicates the point at which the sediment core was retrieved.

### 2.2 Sediment coring and sampling

A slide-hammer gravity corer (Aquatic Research Instruments, Hope, ID, USA) with an inner core-tube diameter of 6.8 cm was used to collect a 26 cm sediment core (referred to as SSBG-21 hereafter) from the lake central area in September 2021. The sediment core was sectioned on-site at 0.25 cm intervals for the first 9.25 cm and at 0.5 cm intervals for the remaining sediment. For each section, 1 cm<sup>3</sup> were immediately taken from the centre of the core, using disposable material to prevent contamination between samples, and stored in small plastic vials for pigment analysis avoiding the exposure of the sediment

to light. The remaining sample was extruded into plastic zip-bags, later freeze-dried, and then stored at 4 °C for later analyses.

### 2.3 Sediment chronology

Freeze-dried subsamples were analysed for <sup>210</sup>Pb, <sup>226</sup>Ra and <sup>137</sup>Cs by direct gamma spectroscopy in the Liverpool University

Environmental Radioactivity Laboratory. A total of 18 subsamples between the top of the core and the 14 cm depth sediment were analysed. Sediment ages were estimated from unsupported <sup>210</sup>Pb activities using the constant rate of supply (CRS) and constant initial concentration (CIC) models (Appleby and Oldfield, 1978). <sup>137</sup>Cs was used as an additional dating marker, representing fallout emitted during thermonuclear bomb testing (with its peak in 1963) and nuclear accidents such as the Chernobyl accident in 1986 or the Fukushima accident in 2011.

### 150 2.4 Carotenoid pigment analysis

A total of 51 samples were used for pigment analyses, using freeze-dried sediments stored in the dark. The pigments were extracted using 90% acetone with a probe sonicator (Sonopuls GM70 Delft, The Netherlands) (50 W, 2 min). The extract was centrifuged (4 min at 3000 rpm, 4°C), filtered through a Whatman ANODISC 25 (0.1 mm) filter and analyzed with ultrahigh-performance liquid chromatography (UHPLC) following a modification of the method described by Buchaca and Catalan (2007). Pigments were identified by comparison with a library of pigment spectra obtained from extracts of pure algal cultures from the Culture Collection of Algae and Protozoa (CCAP, Oban, Scotland, UK) and pigment standards (DHI Water and Environment, Hørsholm, Denmark). Pigments with a high taxonomic affinity were selected, including  $\beta$ -carotene (all groups), scytonemin (UV-screen pigment in cyanobacteria), alloxanthin (cryptophytes), aphanizophyll (N<sub>2</sub>-fixing cyanobacteria), canthaxanthin and echinenone (mainly cyanobacteria and zooplankton), astaxanthin (zooplankton and some chlorophytes), diadinoxanthin (mainly dinoflagellates), diatoxanthin (diatoms), zeaxanthin (chlorophytes and cyanobacteria) and lutein (chlorophytes). We based our interpretation on carotenoid profiles instead of phorbins because there is a higher loss of phorbins than of carotenoids to undetectable colourless compounds when pigments are deposited and buried (Buchaca and Catalan, 2008). Values for  $\beta$ -carotene were expressed as  $\mu$  pigments (excluding  $\beta$ -carotene). The molecular weights of the different pigments were obtained from Jeffrey et al. (1997).

### 2.5 Biological and paleo-environmental data analysis

For elemental analyses of carbon and nitrogen content of the organic matter, a sediment fraction of  $\sim 1$  g of wet weight (WW) sediment was freeze-dried and treated with 1M hydrochloric acid overnight at 50 °C to remove carbonates. The elemental analysis was performed using a Thermo Scientific Flash 2000 coupled to a mass spectrometer at the Scientific Instrumentation Center of the University of Granada (CIC) determining the content of sediment Carbon/Nitrogen (C/N) ratio

calculated from the mass data and expressed as atomic ratios. The carbon to nitrogen (C/N) ratio is commonly used to identify organic matter sources in lacustrine sediments (Meyers and Ishiwatari, 1993). Algal-derived organic matter typically has C/N ratios between 4 and 10, while terrestrial vascular plants show C/N ratios above 20. C/N values between 10 and 20 suggest a mix of both sources (Meyers and Teranes, 2001).

Nitrogen isotope (δ<sup>15</sup>N) composition was determined from the bulk sediment organic matter. It was performed on the remaining carbonate-free sediment samples using the elemental analyser Flash HT Plus coupled to IRMS DELTA V Advantage at the CIC. The δ<sup>15</sup>N values were calibrated to atmospheric N<sub>2</sub>. The complexity of the nitrogen biogeochemical cycle complicates the interpretation of sedimentary δ<sup>15</sup>N (Meyers and Teranes, 2001), this topic will be discussed in a later section.

Freeze-dried sediment samples were analysed for inferred total organic carbon in water following the methodology described in Meyer-Jacob et al. (2017) and sedimentary chlorophyll-*a* including its derivatives (VRS-inferred chl-*a*) according to Michelutti et al. (2010) at the Paleoecological Environmental Assessment and Research Laboratory (PEARL), Queen's University (Canada). Chlorophyll-*a* results include chlorophyll isomers and its major derivatives (pheophytin and pheophorbide) (Michelutti et al., 2010; Michelutti and Smol, 2016). In our case, water TOC is considered virtually equivalent to DOC in lake water due to the dominant contribution of DOC in the Sierra Nevada lake's organic carbon pool (Mladenov et al., 2008), a pattern also observed in lakes with similar characteristics (Meyer-Jacob et al., 2019). From this point onward, analyzed TOC in water will be referred to as DOCw.

### 2.6 Climate and atmospheric data input

The climate data that was used for statistical analysis to explore relationships with the pigment matrix was obtained from Sigro et al. (2024), who developed a high quality daily climate data base specifically for the Sierra Nevada Mountains (<a href="http://www.c3.urv.cat/climadata.php">http://www.c3.urv.cat/climadata.php</a>). We used the mean annual temperature anomaly (MATA) and annual precipitation anomaly (APA). Seasonal mean temperature anomaly (MTA)- springMTA, summerMTA, fallMTA, and winterMTA-, and seasonal precipitation anomaly (PA)- springPA, summerPA, fallPA, and winterPA-, were used to correlate with the annual climate data through Pearson's correlation analysis. Climate data was available for the period ~1894-2020 yr CE.

The intensity of Saharan dust emission and transport has been linked to wNAO (winter North Atlantic Oscillation; Moulin et al., 1997) and to the Sahel drought (Chiapello et al., 2005; Moulin and Chiapello, 2004). Furthermore, Jiménez et al. (2018) demonstrated that the SPI and wNAO indices can be utilised as predictors of the transport and intensity of Saharan dust events in Sierra Nevada, reflecting atmospheric P and Ca deposition trends in this region. These indices exhibited strong correlations with the zirconium-to-aluminium (Zr/Al) ratio—a proxy for Saharan dust deposition—measured in a sediment core from one of the Sierra Nevada lakes, as well as with calcium (Ca) concentrations in an ice core from the French Alps, which are indicative of Saharan dust events (Jiménez-Espejo et al., 2014; Preunkert and Legrand, 2013). Consequently, both indices were employed for this purpose in the present study. The wNAO index (DJFM) defined by Hurrell (1995) is based on the difference of normalized sea level pressure between the Azores High (Lisbon, Portugal) and the Icelandic Low

(Stykkisholmur, Iceland) stations. Hurrell's wNAO index dataset extends back to 1864 on a monthly mean basis (<a href="https://climatedataguide.ucar.edu/climate-data/hurrell-north-atlantic-oscillation-nao-index-station-based">https://climatedataguide.ucar.edu/climate-data/hurrell-north-atlantic-oscillation-nao-index-station-based</a>). The Sahel precipitation index (SPI), extending back to 1900 yr CE, provided by the University of Washington and the Joint Institute for the Study of the Atmosphere and Ocean (<a href="http://research.jisao.washington.edu/data/sahel/">http://research.jisao.washington.edu/data/sahel/</a>), provided a standardized rainfall index data for the Sahelian zone of northern Africa.

The annually resolved climate and dust metrics were averaged over the period of accumulation for each dated interval, thereby integrating the instrumental data with the paleolimnological data (Jiménez et al., 2018). Final weighted data are only available for the period 1905-2020 yr CE (corresponding to samples 1-22).

### 2.7 Statistical analysis

Before statistical analyses were applied, data were transformed to reduce the asymmetry of the distributions and to standardise them. The pigment variables (expressed as relative abundance) were square root transformed (Hellinger transformation). The  $\beta$ -carotene pigment was excluded from the pigment matrix on the basis that it was considered to be a proxy for total algal biomass (Buchaca et al., 2019; Zhang et al., 2019). The explanatory biological and paleo-environmental variables (including  $\beta$ -carotene) were logarithmically transformed followed by a relative scale transformation (y'=y<sub>i</sub>/y<sub>max</sub>). Finally, the climate and dust metric variables (MATA, APA, wNAO and SPI) were relative-scale transformed. All transformed variables met the assumptions of normality and homoscedasticity for the application of parametric statistics. The  $\delta$ <sup>15</sup>N values were used untransformed as these met the conditions of normality and homoscedasticity.

Changes in algal communities were explained using the matrix of pigment values due to the association of individual pigments with one or a few algal groups. Pigment zones were identified through cluster analysis using constrained incremental sum of squares (CONISS), square root transformation of percentage data and chord distance as the dissimilarity coefficient using the program Tilia Graph View (TGView), version v3.0.1 (Grimm, 2011), with the number of important zones determined by the broken stick model (Bennett, 1996).

Principal component analysis (PCA) was performed on the matrix of individual pigments to summarize the major patterns of variability in pigment assemblages into a few axes. Separate PCA analyses were conducted for two time periods: ~430 years (from ~1600 yr CE to the present) and ~115 years ago (from ~1850 yr CE to the present). PCA was also performed on the matrix of atmospheric and annual climate variables (wNAO index, SPI, MATA and APA) to highlight the main patterns of variation, focusing on the 115-year period. Previously, detrended correspondence analysis (DCA) results (gradient length 

were then used to establish the significance of each variable and the percentage of the variance explained, with a significance level set at p < 0.05. Significant environmental variables were incorporated into the final model and subjected to dbRDA once again.

Generalized linear models (GLM) with forward/backward selection were performed to explore potential annual climate and atmospheric drivers of the main variables reflecting primary producer abundance and organic matter composition (VRS-inferred Chl-*a*, β-carotene and C/N ratio). In the GLM analyses, the optimum model was selected using Akaike's information criterion adjusted for sample size (AICc; Burnham and Anderson, 2004). Models with an AICc difference of less than 2 compared to the lowest AICc were considered the best models and statistically equivalent. The contribution of each variable to the final model was determined by assessing its significance and percentage of variance explained. Residuals of the final models were examined to check for normality of data and absence of over-dispersion.

GLM and db-RDA statistical analyses were conducted solely for the period during which atmospheric and climate data were available (~1905-2020 yr CE; 22 samples). The explanatory variables used in both analyses included annual temperature and precipitation data, as well as SPI and wNAO. Seasonal temperature and precipitation data were not included in these analyses to reduce the number of explanatory variables. Pearson correlations between annual and seasonal climate variables were performed.

Ordination and model selection analyses were performed using the vegan (Oksanen, J., 2022) and the MuMIn (Multi-Model Inference; Bartón, 2025) packages for the R software environment, respectively. In all the analyses the variance inflation factor (VIF) was used to eliminate highly correlated variables before applying analyses. All the explanatory variables that yielded VIFs >5 were eliminated in the analysis due to the high degree of collinearity.

### 3 Results

### 3.1 Geochronology

The total activity of <sup>210</sup>Pb (Fig. 2A) reached levels close to equilibrium with the supporting <sup>226</sup>Ra at a depth of 6-7 cm in the core SSBG-21. According to the CRS and CIC models (Appleby and Oldfield, 1978), the concentrations of unsupported <sup>210</sup>Pb below 1 cm decrease exponentially with depth (unshown data). This indicates a relatively uniform accumulation of sediments over much of the time period spanned by the core. The relatively uniform concentrations observed in the top 1 cm may be attributed to a recent slight increase in sedimentation rates. Excluding the top 1 cm, the average sedimentation rate is calculated to be 0.0098 ± 0.008 g cm<sup>-2</sup> y<sup>-1</sup> (or 0.041 cm y<sup>-1</sup>) (Fig. 2B). Additionally, <sup>137</sup>Cs concentrations showed a distinct peak at a depth of 1.5-1.75 cm (Fig. 2A), likely reflecting fallout from the Chernobyl accident in 1986. The reliable dating period for the core extends from 6.63 cm to the present, covering approximately ~1850 to 2020 yr CE, with an estimated mean temporal resolution of 5 years per sampling interval. For sediment layers from the bottom of the core up to 6.63 cm, sediment ages were estimated through extrapolation (with an average of approximately 14 years per sampling interval). Thus, it is important to interpret these ages with caution.

Figure 2: A) Radiometric chronology showing <sup>210</sup>Pb (black circle) and <sup>137</sup>Cs (gray circle) activity (bq kg<sup>-1</sup> dried sediment) from sediment core SSBG-21 extracted from Borreguil Lake. B) <sup>210</sup>Pb-estimated age (using the constant rate of supply model) versus core depth (black line) and the associated errors (grey lines).

### 3.2 Trends in pigment composition

Throughout the whole core, the pigments associated with Chlorophyceae (lutein and zeaxanthin) and those associated with diatoms (diatoxanthin) were predominant (Fig. 3). Both lutein and zeaxanthin were attributed to have an origin from green algae, given that their relative abundance profiles change in parallel. Diatoxanthin had an overall increasing trend from the bottom of the core to ~1970 yr CE, after which it declined towards the top of the core. Conversely, lutein and zeaxanthin showed an opposite trend to diatoxanthin.

Figure 3: Relative abundance diagrams of the sedimentary pigments recorded in the sediment core SSBG-21 in Borreguil Lake. The result of a cluster analysis of pigment assemblages data using constrained incremental sum of squares (CONISS) is shown. The black lines represent the main zonation identified by the broken stick model. The data are plotted against the sediment depth (cm, primary y-axis) and on age (yr CE, secondary y-axis). Dates prior to 1850 should be interpreted with caution, as the age provided is beyond the confidence dating provided by stable radioisotopes.

Cyanobacteria were the second most abundant algal group, with echinenone, aphanizophyll and scytonemin as indicator pigments (Fig. 3). Significant echinenone production started ca. 1750 yr CE, maintaining a stable abundance of about 5% until the top of the core. Aphanizophyll appeared sporadically between ca. 1750-1800 yr CE, then disappeared around 1850 yr CE, after which it increased from 1-2% to 10% towards the core top (Fig. 3). Scytonemin, on the other hand, was recorded only occasionally from 1780-1850 yr CE. Thereafter, its abundance remained minimal until it rose to approximately 2% by 1960 yr CE, at which point it persisted until the top (Fig. 3).

The least abundant pigments were diadinoxanthin (a marker pigment for Dinophyceae) and alloxanthin (a marker pigment of cryptophytes), which had an abundance of less than 5% throughout (Fig. 3). The appearance of diadinoxanthin was recorded around 1750 yr CE, reaching a maximum of 5% until 1870 yr CE, after which it declined. Alloxanthin first appeared in 1900 yr CE, maintaining a stable abundance of about 2% to the top of the core. In addition to the aforementioned pigments, astaxanthin and canthaxanthin, which are associated with crustacean tissues, were also detected (Fig. 3). The relative

abundances of these pigments were highest from the bottom of the core to  $\sim$ 1750 yr CE, with a progressive decline thereafter to the top of the core.

According to the CONISS analysis the sediment pigment sequence can be divided in five significant zones of change (Fig. 3): at ~1970, ~1843, ~1804, and ~1690 yr CE. The main change took place at ~1843 yr CE and was characterized by the appearance of aphanizophyll, an increase in diatoxanthin and a decrease in canthaxanthin. The second most significant change took place at ~1970 yr CE to the top of the core where there was a decrease in diatoxanthin and diadinoxanthin (diatoms and Dinophyceae, respectively), an abrupt increase in aphanizophyll and scytonemin (cyanobacteria) and alloxanthin (cryptophytes), and a subtle increase in lutein and zeaxanthin (Chlorophyceae).

PCA of the entire core (51 samples) explained 73.91% of the total pigment variation (45.84% for axis 1 and 28.07% for axis 305 2) (Fig. 4A and 5A). Axis 1 showed two periods: a decrease until ~1670 yr CE and an upward trend after that, with some fluctuations. Axis 2 tracked an increasing trend around ~1705-~1780 yr CE and ~1850-~1900 yr CE, followed by a sharp decline after ~1950 yr CE. Axis 1 was linked to echinenone, aphanizophyll, diadinoxanthin, and alloxanthin in the positive region, and Crustacea-related pigments (astaxanthin, canthaxanthin) in the negative region. Axis 2 was associated with zeaxanthin, lutein, and diatoxanthin, with scytonemin showing no clear relationship to either axis.

PCA of the climatic data period (22 samples) explained 64.49% of the total pigment variation (44.01% for axis 1 and 20.48% for axis 2) (Fig. 4B and 5B). Axis 1 showed a general decrease, especially since the ~1960s, while axis 2 displayed oscillatory values. Axis 1 was associated with zeaxanthin, alloxanthin, lutein, and aphanizophyll in the negative region, and diatoxanthin in the positive region. Axis 2 was linked to diadinoxanthin, astaxanthin, and echinenone in the positive region, and canthaxanthin in the negative region. Scytonemin showed no clear relationship to either axis., being positioned in the upper left quadrant.

Figure 4: Evolution of sample scores on PCA axes 1 and 2 from the pigment matrix of sediment core SSBG-21 extracted from Borreguil Lake over (A) the last 430 years; and (B) the last 115 years. Dates prior to 1850 should be interpreted with caution, as the age provided is beyond the confidence interval for radioisotopes dating.

Figure 5: Principal component analysis (PCA) biplots of the pigment assemblages from sediment core SSBG-21 extracted from Borreguil Lake for (A) the last  $\sim$ 430 years; and (B) the last  $\sim$ 115 years. Numbers refer to the sample sites being 1 the most modern sample and 51 and 22 the oldest samples in A and B, respectively.

### 3.3 Instrumental climate data

Figure 6 shows the mean annual air temperature and annual precipitation anomalies in Sierra Nevada for the period 1894-2020 yr CE. The average annual temperature exhibited a significant increase rate of 0.12 °C/decade, with a total increase of 1.56 °C for the 126 years analysed. The warming in MATA was mainly driven by summer and spring temperatures, followed by autumn and winter temperatures (Sigro et al., 2024). This trend was also observed in the correlation analysis, which yielded a correlation factor of 0.97 and 0.94 between MATA, and summer and spring mean temperatures, respectively

(Supplementary Fig. S1). Mean annual (Fig. 6) and seasonal mean temperatures (Supplementary Fig. S2) show negative anomalies prior to the 1940s and there were negative values around the 1970s. The rest of the periods correspond to positive anomalies, especially since ~1995 yr CE where the annual, spring and summer anomalies were higher than in any previous period. The rise in the winter and autumn anomalies were not so marked.

With regard to the precipitation anomalies, a downward trend was observed, particularly during the period 1975–2020 yr CE, which exhibited a significant negative trend of -12.9% per decade in summer precipitation (Fig. 6; Sigró et al., 2024).

Figure 6: Evolution of the climatic variables MATA (mean annual temperature anomalies for air), and APA (annual precipitation anomalies) at the summits of Sierra Nevada for the period 1894-2020, and Saharan dust metrics SPI (Sahel Precipitation Index) and wNAO (winter North Atlantic Oscillation) index. Temperature and precipitation data were obtained from Sigró et al. (2024). The anomalies of the SPI are calculated with respect to 1900 and 2013, and based on June through October averages for each year.

### 3.4 Atmospheric deposition variables

The wNAO index showed a consistent upward trend (positive index values) starting from the mid-1970s yr CE, reaching its peak during the 1980s-1990s (Fig. 6). In contrast, the SPI record indicated a predominantly wet period (positive values) from 1900 yr CE to approximately 1970 yr CE, followed by a relatively stable and dry period (negative values) from around 1970 yr CE to the present, with the lowest values observed during the 1980s-1990s yr CE (Fig. 6).

In the PCA biplot (Fig. 7), atmospheric and annual climate variables are placed in different quadrants. More modern samples are positively associated with higher values of temperature and more positive values of the wNAO index, while they are negatively associated with higher values of the SPI and lower temperature values.

Figure 7: Principal component analysis biplots of atmospheric and annual climate variables for the last 115 years. Numbers refer to the sample sites being 1 the most modern sample and 22 the oldest samples.

### 3.5 Trends in other paleo-environmental data

VRS-inferred Chl-a and  $\beta$ -carotene (variables related to total algal biomass) show similar trends throughout the entire profile (Fig. 8). Both variables have an overall increasing trend towards the present with two distinct periods of higher concentration: a first one between ~1740 and ~1840 yr CE, and a second one from ~1950 yr CE to the present. The C/N ratio demonstrated fluctuations in values from the base of the core to approximately 1970 yr CE, exhibiting a discernible and

consistent decline thereafter. A marked increase in the C/N ratio values occurred between 1740 and 1840 yr CE.  $\delta^{15}$ N values remained relatively stable, exhibiting a slight but constant decrease until approximately 1980 yr CE, at which point it underwent a steep decline (Fig. 8). DOCw values did not show a discernible trend throughout the core, with highly oscillating values. However, two periods of increasing DOCw values can be distinguished: a first one between ~1810 and 1840 yr CE and a second one from ~1975 yr CE onwards.

Figure 8: Evolution of selected paleoenvironmental variables from sediment core SSBG-21 extracted from Borreguil Lake. Dates prior to 1850 should be interpreted with caution, as the age provided is beyond the confidence dating provided by stable radioisotopes. Vertical lines represent the main divisions identified in the CONISS analysis in the pigment assemblages.

From the second half of the 20th century onwards, a more discernible pattern became evident across the variables, coinciding with the period of highest temperature and low precipitation, as well as low SPI and high wNAO values. During this period, the highest values recorded for  $\beta$ -carotene and VRS-inferred Chl-a were observed. Conversely, the C/N ratio and  $\delta^{15}$ N values reached their lowest values for the entire sediment record (Fig. 8).

Aphanizophyll is produced by nitrogen-fixing cyanobacteria and this pigment exhibit a strong Pearson negative correlation with  $\delta^{15}N$  (r = -0.84), and both variables display opposite profiles throughout the entire core (Fig. 9). Of particular note is the period from ~1900 yr CE onwards, where aphanizophyll relative abundance increased at the same time that  $\delta^{15}N$  decreased. During this period, the  $\delta^{15}N$  values reached their lowest point, while those of aphanizophyll reached their highest.

Figure 9: Evolution of the concentration of the Aphanizophyll pigment (marker pigment of  $N_2$ -fixing cyanobacteria) and  $\delta^{15}N$  from the sediment core SSBG-21 taken in Borreguil Lake. Dates prior to 1850 should be interpreted with caution, as the age provided is beyond the confidence interval for radioisotopes dating.

### 3.6 Relationships between pigment data and paleoenvironmental and instrumental climate data over the last 115 years

Prior to all the analyses, VIF>5 analyses were performed on the explanatory variables and no variables were discarded. The resulting dbRDA model produced an R<sup>2</sup> adjusted value of 0.468, with MATA and SPI identified as the primary drivers of pigment composition across the period ~1905-2020 yr CE (Fig. 10; Table 1).

To explore the relationships between VRS-inferred Chl-a,  $\beta$ -carotene and C/N ratios and annual climate and atmospheric variables, GLM analyses were undertaken (Table 2). MATA was selected as the main explanatory variable for VRS-inferred

Chl-a,  $\beta$ -carotene, and C/N ratio, being APA and wNAO secondary explanatory variables for  $\beta$ -carotene and C/N ratio, respectively.

Figure 10: The results of the distance-based Redundancy Analysis (db-RDA) ordination plot showing the relationship between pigment assemblages data from sediment core SSBG-21 and the climate and atmospheric variables. The selected explanatory variables by the db-RDA analysis are shown. MATA (Mean annual air temperature) and SPI (Sahel Precipitation Index as proxy of Saharan dust input). Only the first two db-RDA axes are shown. Sample sites (representing sediment intervals) are also shown, being "1" the most modern interval and "22" the oldest one. See Table 3 for more details of the analysis results.

Table 1: Summary of results from the distance based redundancy analyses (dbRDA) with pigment matrix as response variable and the climate variables as explanatory variables for BG Lake for the period  $\sim$ 1905-2020. Forward/backward-selection was used to select the explanatory variables for the db-RDA, that were scale-relative transformed to standardize to mean variance. Only selected predictor variables for the dbRDA are shown. Adj  $R^2$  = Adjusted  $R^2$ . Significance levels: \*\*\* P < 0.001; \*\* 0.001 < P < 0.01.

|                    | df    | Variance | F      | p-values |
|--------------------|-------|----------|--------|----------|
| MATA               | 1     | 0.787    | 11.626 | 0.001*** |
| SPI                | 1     | 0.412    | 6.091  | 0.004**  |
| Residual           | 19    | 1.29     |        |          |
| Adj R <sup>2</sup> | 0.468 |          |        |          |

Table 2: Summary of results from the model selection analyses (generalized linear models-GLM) predicting the VRS-inferred Chl-a,  $\beta$ -carotene and C/N ratio for BG Lake for the period ~1905-2020. The explanatory variables were relative scale transformed to standardize to mean variance. The best model according to the Akaike's information criterion (AICc) values is shown. Predictor variables for the analyses include: MAPA, Sierra Nevada mean annual air temperature anomaly; APA, Sierra Nevada annual precipitation anomaly; wNAO, winter North Atlantic Oscillation index and SPI, Sahel precipitation index (proxy for Saharan dust input). Adj  $R^2$ , adjusted  $R^2$ . \*\*\* Significance level p < 0.001; \* Significance level 0.01 < p < 0.05.

| Response variable | Explanatory variables | Adj R <sup>2</sup> | $\chi^2 = LRChisq$ | df | p-values                                                 |
|-------------------|-----------------------|--------------------|--------------------|----|----------------------------------------------------------|
| Chl-a             | MATA                  | 0.419              | 16.125             |    | 5.93 ·10-5 ***                                           |
| β-carotene        | MATA<br>APA           | 0.185              | 3.920<br>3.355     |    | 4.77· 10 <sup>-2</sup> * 6.70· 10 <sup>-2</sup> *        |
| C/N               | MATA<br>wNAO          | 0.539              | 21.406<br>4.563    | 1  | 3.72 · 10 <sup>-6</sup> ***<br>3.27 · 10 <sup>-2</sup> * |

### 4 Discussion

In this study, we have employed different proxies (algal subfossil pigment assemblages and other paleoenvironmental biological variables) derived from the Borreguil Lake sediment core with the objective of gaining a deeper understanding of past regional climate-driven changes for Sierra Nevada lakes and alpine lakes in general. The algal community exhibited notable changes, both in the algal biomass and assemblage composition throughout the entire core (1600 yr CE-present), which were mainly driven by climate and atmospheric variables. The most pronounced changes, both in algal biomass and assemblages, occurred during the latter half of the twentieth century coinciding with a period of marked rise in temperature and Saharan dust deposition, and a decreasing trend in the precipitation.

### 4.1 Has the primary production and/or the algal biomass of Borreguil Lake increased during the period ~1600-2020 yr CE? What variables have likely contributed to this trend?

The combined analysis of VRS-inferred Chl-*a*, β-carotene and C/N ratios enabled us to assess the long-term changes in primary production and/or algal biomass throughout the core. Sedimentary VRS-inferred Chl-*a* values varied between 0.04 and 0.1 mg g<sup>-1</sup> DW, which is consistent with the findings reported for six lakes in Sierra Nevada (Jiménez et al., 2018), and other alpine lakes in North America and Europe (e.g. Kang et al., 2019). C/N values (10.6 to 16.8) in our sediment record are comparable to those in other alpine and subalpine lakes in North America (Oleksy et al., 2020; Spaulding et al., 2015) and in the Pyrenees (Vicente de Vera García et al., 2023), and align with current and historical data from other Sierra Nevada lakes (Jiménez et al., 2019; Pulido-Villena et al., 2005). The C/N ratio analysis reveals the main sources of organic matter was a mixture of algal and terrestrial organic matter throughout the record. This is expected due to the lake's small size, with partial coverage of its catchment by alpine meadows.

VRS-inferred Chl-*a* and β-carotene, indicators of algal biomass, show similar trends throughout the core. From the bottom of the core to around 1750 yr CE, both remained stable and low, reaching their lowest levels. This low biomass period may be linked to low temperatures during the Little Ice Age (LIA), as shown by Jiménez-Moreno et al. (2023) for Río Seco Lake. Between ~1750–1800 yr CE, Chl-*a* and β-carotene peaked, even though this interval coincided with the minimum temperatures of the Little Ice Age in the Sierra Nevada (Jiménez-Moreno et al., 2023). A decline in crustacean-associated pigments (astaxanthin, canthaxanthin) may explain this biomass increase. Morales-Baquero et al. (2006) proposed that shorter ice-free periods in colder years limit crustacean populations, favouring algal species. Consequently, diatoms and cyanobacteria likely thrived, as indicated by increased diatoxanthin, diadinoxanthin, echinenone, and aphanizophyll.

Following the LIA (1800-1840 yr CE), a temperature peak at the Sierra Nevada summit was recorded (García-Alix et al., 2020; Jiménez-Moreno et al., 2023). This period saw a shift in pigment composition, with declining diatoms and increasing green algae, as indicated by CONISS and PCA axis 1. The first major algal biomass increase (~1800 yr CE) likely resulted from post-LIA warming, consistent with trends in other alpine lakes (Hu et al., 2014; Huo et al., 2022; Lami et al., 2010; Oleksy et al., 2020). The C/N peak (~1840 yr CE), shortly after the LIA, suggests an influx of external organic matter, potentially driven by thaw-induced transport.

The main changes in VRS-inferred Chl-a and  $\beta$ -carotene occurred from ~1920 yr CE onwards, when a striking increase in both variables was observed. According to GLM analysis (Table 2), climate factors—particularly rising temperatures—were identified as the main drivers of these changes, as well as of the variation in the C/N ratio, over the period 1905–2020 yr CE.

Changes in temperature and precipitation can also clearly affect algal biomass. In alpine lakes, higher temperatures and lower precipitation may extend the ice-free period (Anderson et al., 1996; Rogora et al., 2018), increasing the growing season and resulting in a higher accumulation of annual algal biomass. Longer ice-free seasons in alpine lakes boost light availability, water temperature, and solute inputs through increased snowmelt and weathering (Preston et al., 2016;

Sommaruga-Wögrath et al., 1997). These phenomena can enhance biological production (Douglas and Smol, 2010; López-450 Merino et al., 2011), leading to higher algal biomass. Additionally, the extended growing season allows algal populations to accumulate biomass over time. Previous studies in six high mountain lakes of Sierra Nevada (Jiménez et al., 2018) attributed the increase in chlorophyll-*a* during the 20th century to the lengthening of the growing season, among other factors. In Arctic and alpine lakes, recent climate-driven shifts in primary producer assemblages and ecosystem production underscore the duration of ice cover as a key factor in the response of high-altitude and high-latitude lakes to climate change (Adrian et al., 2009; Rühland et al., 2008, 2015).

The second period of increased algal biomass occurred from the 1960s yr CE to the present, and coincided with a decrease in C/N values, indicating an increased contribution from algae to the system, with one of the main changes in the pigment assemblage, and with a great turnover in the pigment assemblages (PCA axis 1; Fig. 4). Previous studies in high mountain lakes of Sierra Nevada (Jiménez et al., 2018; Pérez-Martínez et al., 2020b) and other alpine regions (Oleksy et al., 2022; Wolfe et al., 2003) have attributed increased algal biomass and/or productivity to a synergistic interaction between elevated temperatures and nutrient enrichment. Since the 1960-70s yr CE, there has been a notable increase in temperature values, particularly in the mean temperatures of the summer and spring months, in the Sierra Nevada region. Conversely, there has been a decline in precipitation levels during the summer months (Sigro et al., 2024). Both the temperature increase and precipitation decrease might have enhanced the algal biomass by the mechanisms previously detailed in the above paragraph.

The combined effects of reduced summer precipitation, higher temperatures, and increased Saharan dust deposition likely contributed to greater algal biomass through nutrient evapo-concentration and atmospheric P deposition from dust. Since the 1950s yr CE, the western Mediterranean has experienced an increase in the frequency of dust events (Salvador et al., 2022), with the Sierra Nevada showing enhanced dust deposition particularly since the 1970s yr CE. Longer ice-free periods may have further amplified dust exposure, boosting nutrient availability and promoting algal growth (Korbee et al., 2012; Saros et al., 2005; Yang et al., 1996). Saharan dust inputs have been shown to influence the structure and composition of biological communities in Sierra Nevada lakes. For example, calcium deposition affects diatom and cladoceran communities (Jiménez et al., 2018; Pérez-Martínez et al., 2020b), while phosphorus inputs enhance algal biomass, leading to higher chlorophyll-a concentrations (Carrillo et al., 1990; Jiménez et al., 2018; Morales-Baquero et al., 2006; Pulido-Villena et al., 2006). Similar Saharan dust fertilization effects have been observed in Pyrenean (Camarero and Catalán, 2012) and United States lakes (Brahney et al., 2015b; Scholz and Brahney, 2022). Recent results from Borreguil Lake show that, in 2022 yr CE, following a massive influx of Saharan dust in Sierra Nevada, there was a significant increase in algal density and a shift in the lake's algal composition, with a marked increase in the percentage of cyanobacteria (Supplementary Fig. S3).

Since the early 1900s yr CE, anthropogenic nitrogen deposition has become increasingly evident in lake sediment cores from high-latitude regions of the Northern Hemisphere (Holtgrieve et al., 2011), particularly during the latter half of the 20th

century (around the 1970s), coinciding with the "Great Acceleration" of global environmental change (Steffen et al., 2007). This reflects the growing influence of nitrogen deposition, mainly from agriculture and fossil fuel combustion, which can significantly drive changes in remote lake ecosystems (Bergström and Jansson, 2006). However, nitrogen deposition in Sierra Nevada is lower than in other Mediterranean regions (Morales-Baquero et al., 2006, 2013) and heavily industrialized areas of Central Europe (Holland et al., 2005), with a mean total nitrogen deposition (dry and wet) of 115.2 mg m<sup>-2</sup> d<sup>-1</sup> between 2000-2002 yr CE. Thus, the observed increase in algal biomass in our lake during the study period is much more likely due to a combined effect of climate warming and increased phosphorus inputs.

## 4.2 Have there been changes in the composition of the algal community of Borreguil Lake in the period ~1600-2020 yr CE? What variables have influenced the algal succession?

The planktonic algal community of Borreguil Lake currently consists of four classes, dominated by Chlorophyceae and Cyanobacteria, followed by Chrysophyceae and Cryptophyceae (Supplementary Fig. S3). Chlorophyceae is the most diverse group, with 17 taxa. Most of the diatoms are periphytic species.

Buchaca and Catalan (2024) highlighted that calcium (Ca) and total phosphorus (TP) concentrations can be important determinants of phytoplankton composition in high-mountain lakes. Below 4 mg L<sup>-1</sup> Ca, such as in our study lake (Jiménez et al., 2018), chlorophytes and chrysophytes dominate. In contrast, when TP levels fall below 5 μg L<sup>-1</sup>, chrysophytes are expected to prevail (Buchaca and Catalan, 2024). In Borreguil Lake, TP ranges from 13 to 27 μg L<sup>-1</sup> (Jiménez et al., 2018), which explains the low abundance of chrysophytes. Yet, the extent to which this relationship applies locally remains to be fully established.

Although chrysophytes are currently part of the planktonic community (Supplementary Fig. S3), their historical presence may be underestimated in the sedimentary record due to pigment preservation issues. Fucoxanthin, a marker pigment for both chrysophytes and diatoms, was detected in surface sediments but was excluded from downcore analysis because of its low molecular stability (Buchaca and Catalan, 2007; Leavitt and Hodgson, 2001). Furthermore, in high-altitude shallow lakes, fucoxanthin is often replaced by more stable pigments such as diatoxanthin in deeper layers, complicating the long-term tracking of chrysophyte and diatom abundance (Buchaca and Catalan, 2008). Thus, the lack of fucoxanthin in older sediment layers should not be interpreted as definitive evidence for the absence of chrysophytes in historical phytoplankton communities.

In the lakes of the Sierra Nevada, the planktonic community coexists with three additional algal communities (Cuesta Linares et al., 2003; Sánchez Castillo and Morales Torres, 1980): (1) a bog-associated algal community comprising Chlorophyceae (particularly Zygnemataceae and Desmidiaceae), along with diatoms and cyanobacteria, some of which are nitrogen-fixing (such as *Nostoc*, *Anabaena*, and *Cylindrospermum*); (2) epilithic biofilms primarily made up of

cyanobacteria and diatoms and epipelic diatoms; and (3) algal mats predominantly composed of filamentous Zygnemataceae and cyanobacteria growing at the littoral and/or bottom of the lake.

Our sedimentary pigment analyses indicate that the algal community in Borreguil Lake was primarily composed of diatoms and Chlorophytes over the last ~430 years. Overall, the changes in the relative abundance of pigments can be summarized from the bottom to the top of the core as an increase in cyanobacteria (indicated by aphanizophyll and scytonemin), green algae (indicated by lutein and zeaxanthin), and cryptophytes (indicated by alloxanthin), at the expense of diatoms (diatoxanthin). Concurrently, there was a decrease in canthaxanthin, astaxanthin, and diadinoxanthin.

### 4.2.1 Period ~1600-1750 yr CE

From the core bottom to ~1750 yr CE, pigments from chlorophytes, diatoms, and Crustacea (canthaxanthin, astaxanthin) dominated. After ~1750 yr CE, crustacean pigments declined, remaining below 5% thereafter. The decline in crustacean abundance coincided with the LIA's minimum temperatures in the Sierra Nevada (1750–1800 yr CE; Jiménez-Moreno et al., 2023) and colder conditions and shorter ice-free periods may have hindered zooplankton communities (Morales-Baquero et al., 2006). The relative abundance of echinenone remained relatively stable throughout the core since its appearance at ~1760 yr CE. The occurrence of echinenone at ~1760 yr CE during a cold period is unexpected, as this cyanobacterial pigment is typically associated with high temperatures. Picoplanktonic cyanobacteria, such as Synechococcales and Chroococcales—common in alpine (Buchaca and Catalan, 2024; Callieri, 2008) and Sierra Nevada lakes (Jiménez, Laura et al., 2015)—may explain its presence. This is further supported by the decline in crustacean pigments (astaxanthin, canthaxanthin) at around 1750, suggesting reduced predation on edible planktonic cyanobacteria (Jiménez et al., 2015). Morales-Baquero et al. (2006) showed that limited zooplankton growth in a Sierra Nevada lake during cold years weakened top-down control, increasing algal biomass compared to warmer periods.

### 4.2.2 Period ~1800-1970 yr CE

Between 1800 and 1840 yr CE, crustacean pigments, especially canthaxanthin, increased, likely due to rising temperatures. However, from 1840 yr CE onward, astaxanthin and especially canthaxanthin declined. While no direct data explain this decrease, shifts in trophic interactions, possibly due to the expansion of predatory invertebrates (e.g., Corixidae, Dytiscidae, odonate larvae), may have impacted zoobenthos and/or zooplankton. The decline in astaxanthin and canthaxanthin may also reflect a shift in crustacean composition, with fewer pigmented species and/or more less-pigmented ones. However, the available data do not permit further exploration of these hypotheses.

The main change in pigment composition took place at ~1840 yr CE, as indicated by the CONISS analysis and the PCA axis 1 (Figs. 3 and 4A). This change in pigment composition was characterized by the first appearance of aphanizophyll and a decrease in canthaxanthin and astaxanthin. This pigment shift aligns with global aquatic ecosystem restructuring since the mid-19th century, driven by intensified human impact, including nutrient enrichment (Dubois et al., 2018). However, direct

anthropogenic influences did not occur for Sierra Nevada at that time (Anderson et al., 2011; García-Alix et al., 2013). This change may instead be explained by a summer temperature peak in Sierra Nevada (based on chironomids and leaf waxes) between 1800 and 1840 yr CE (García-Alix et al., 2020; Jiménez-Moreno et al., 2023). Warming could explain the increased relative abundance of cyanobacteria, which thrive at higher temperatures (Havens and Paerl, 2015).

The period from 1840 to 1970 yr CE was characterized by the absence of significant shifts, with the algal community remaining essentially unchanged. This stability between 1840 and 1970 may be explained by the relatively moderate temperature increase during that period; although temperatures rose by over 2 °C from 1870 to the present (7.8 to 10.6 °C), the most pronounced warming occurred after 1950 (from 8.7 to 10.6 °C), according to chironomid-based reconstructions (Jiménez-Moreno et al., 2023). Similarly, Sigro et al. (2024) report that warming in the Sierra Nevada began around 1930 but intensified notably after 1975, potentially explaining the marked ecological changes observed since the 1970s.

### 4.2.3 Period ~1970-2020 yr CE

From ~1970 yr CE onwards, significant shifts in pigment assemblages were observed, characterized by increases in cyanobacteria (aphanizophyll, scytonemin), cryptophytes (alloxanthin), and green algae (lutein, zeaxanthin), while diatoms (diatoxanthin) decreased. This shift coincided with higher temperatures, decreased precipitation, increased Saharan dust inputs (Fig. 6), and a greater algal contribution to sediment (i.e. lower C/N ratio). Algal biomass indicators (VRS-inferred chl-*a* and β-carotene) showed a positive trend from the 1930s yr CE, with a slight and punctual decline around the 1970s yr CE. These changes in pigments, Chl-*a*, β-carotene, and C/N ratios were primarily driven by climate and atmospheric events, especially temperature, as indicated by the dbRDA (Table 1). Data from ~1970-2020 yr CE show unprecedented relative abundances of cyanobacteria and cryptophytes, along with high algal biomass and increased algal contribution (C/N ratios). These trends are unparalleled in the sediment core record. Previous studies in Sierra Nevada lakes also report increased biomass since ~1970 yr CE, linked to rising temperatures and intensified Saharan P deposition (Jiménez et al., 2018). However, no studies had analyzed so far long-term changes in the algal composition.

The shift in algal assemblage composition around 1970 yr CE in Borreguil Lake is mainly attributed to changes in temperature, precipitation, nutrient availability, and UV radiation, through mechanisms that differ among algal groups and are elaborated upon in the following sections.

### 4.2.3.1 Increase in the relative abundance of cryptophytes, cyanobacteria and chlorophytes

The increase in cryptophytes and chlorophytes in our study lake aligns with the rise of alloxanthin and green algae marker pigments in the Alps during the same period (Hofmann et al., 2021). This trend is linked to rising temperatures and water column stabilization, with both cryptophytes and green algae serving as indicators of thermal stabilization (Wolfe et al., 2003). The temperature increase since the 1970s yr CE likely favoured taxa with higher thermal optima, such as chlorophytes (Barone and Naselli-Flores, 2003; Elmslie et al., 2020; Florian et al., 2015) and cyanobacteria (Havens and Paerl, 2015), over diatoms. Additionally, phosphorus enrichment in the lake during this period (Jiménez et al., 2018) may

have enhanced the performance of chlorophytes, cyanobacteria, and cryptophytes (Buchaca and Catalan, 2024; Oleksy et al., 2020; Zufiaurre et al., 2021). For example, Oleksy et al. (2020) demonstrated that, in high mountain lakes, the rise in chlorophytes and cyanobacteria was driven by increased phosphorus and nitrogen, with climate warming and phosphorus being the main explanatory factors. In addition to phosphorus and nitrogen, calcium (Ca) transported by Saharan dust is also a key element influencing algal community composition. For example, Brahney et al. (2015) suggest that phosphorus associated with carbonate dust deposition promotes cyanobacterial growth in an alpine lake, while diatom growth is not similarly favored. In laboratory experiments González-Olalla and Brahney (2025) found that the combination of temperature increase and carbonate-rich dust enrichment led to greater cyanobacteria growth in an alpine lake. Moreover, in the Pyrenees, diatoms were partially replaced by cyanobacteria at approximately 1.2 mg L<sup>-1</sup> of calcium (Buchaca and Catalan, 2024). In our study lake, calcium concentrations (0.8–1.1 mg L<sup>-1</sup>; Jiménez et al., 2018) are near this threshold, allowing both diatoms and cyanobacteria to coexist with fluctuating proportions. However, the relationship between Ca, TP, and phytoplankton may not be consistent across all high mountain lakes. In Sierra Nevada lakes, for example, Saharan calcium inputs are particularly significant and may promote cyanobacterial growth and persistence. The large influx of Saharan dust in 2022, for instance, led to a marked increase in the proportion of cyanobacteria in these lakes (Supplementary Fig. 3). Our dbRDA results reveal that mean annual temperature and phosphorus-rich Saharan dust deposition were key drivers of pigment assemblage changes, with temperature being the primary factor. In contrast, Taranu et al. (2015) found that the increase in cyanobacteria in alpine and lowland lakes over the past 200 years was mainly due to rising nutrient levels (N and P), with temperature playing a secondary role. This discrepancy in the relative importance of temperature can be attributed to geographic differences in study locations. Taranu et al. (2015) focused on lowland lakes, where eutrophication tends to have a stronger impact than in remote alpine regions.

### 4.2.3.2 Increase in the relative abundance of N2-fixing cyanobacteria

Climate- and Saharan-driven nutrient changes may have caused a stoichiometric imbalance, especially since the 1970s yr CE, with an increase in potentially  $N_2$ -fixing cyanobacteria (aphanizophyll), coinciding with a sharp decrease in  $\delta^{15}N$  values (Fig. 9).  $\delta^{15}N$  in organic matter can be depleted for two main reasons (Meyers and Teranes, 2001). First, increased atmospheric nitrogen (DIN) inputs, with a low isotopic signature, can lower  $\delta^{15}N$  values. Anthropogenic nitrogen, typically depleted in  $\delta^{15}N$  compared to natural sources (Talbot, 2001), can also reduce  $\delta^{15}N$  when deposited in aquatic systems. Second, an increase in  $N_2$ -fixing cyanobacteria can reduce  $\delta^{15}N$ , as the isotopic signature of  $N_2$  is also low. In our study site, the significant decline in  $\delta^{15}N$  values are similar to the abrupt decreases observed in high-altitude lakes across America (Holtgrieve et al., 2011; Spaulding et al., 2015; Wolfe et al., 2003), the Alps (Hofmann et al., 2021), and Arctic regions (Florian et al., 2015) primarily attributed to increased atmospheric nitrogen deposition. However, the sharp decrease in  $\delta^{15}N$  values at ~1990 yr CE contrasts with the observed decline in  $\delta^{15}N$  values in other lakes from 1970 and with the decline in nitrogen deposition in Europe after 1980 (Engardt et al., 2017). Moreover, in the Sierra Nevada region, nitrogen deposition is relatively low compared to other Mediterranean areas (Morales-Baquero et al., 2006; Morales-Baquero and Pérez-Martínez,

2016) and heavily industrialized regions in Central Europe (Holland et al., 2005). On the other hand, the depletion of  $\delta^{15}N$ due to N<sub>2</sub> fixation by cyanobacteria, driven by low nitrogen availability in Borreguil Lake, is supported by several factors. First, since the mid-20th century, and especially since the 1980s yr CE, the deposition of P-rich Saharan dust has increased in Sierra Nevada lakes (Fig. 6; Jiménez et al., 2018). Second, several significant climate events occurred during this period, including a rise in temperatures starting in the 1970s (Fig. 6), a decrease in summer precipitation (Sigro et al., 2024; Fig. 6), and the onset of recurrent summer droughts in Sierra Nevada since the 1980s yr CE (Pardo-Igúzquiza et al., 2024). Reduced precipitation levels would lead to a marked decline in wet deposition, which is the primary pathway for atmospheric nitrogen input into Sierra Nevada lakes (Castellano-Hinojosa et al., 2017; Morales-Baquero et al., 2013). Additionally, lower precipitation reduces nitrogen transport from the catchment to lakes, further decreasing nitrogen input (Morales-Baquero et al., 1999). These factors, combined with the high phosphorus-rich Saharan dust deposition over the past 50 years—especially in spring and summer—must be considered. Dry deposition predominates during this period, driven by a strong thermal anticyclone system in northern Africa (Escudero et al., 2005), a pattern also observed in the Sierra Nevada (Morales-Baquero and Pérez-Martínez, 2016). Since this input occurs mainly during the ice-free period, it likely has a greater impact on lake biota compared to nitrogen deposition, which is more concentrated during the ice-cover period. Together, these factors may have led to a reduction in the N/P ratio, decreasing nitrogen availability. This, in turn, could have favoured the growth of N<sub>2</sub>fixing cyanobacteria, causing a decline in δ<sup>15</sup>N values linked to N<sub>2</sub> fixation (Meyers and Teranes, 2001). For example, dust deposition can modify nutrient dynamics in freshwater ecosystems by altering nitrogen-to-phosphorus (N:P) ratios, which in turn can reshape community composition and stimulate both primary production and phytoplankton growth. These nutrientdriven responses have been observed in studies in Pyrenees and Sierra Nevada (Camarero and Catalán, 2012; González-Olalla et al., 2018) as well as across the western United States (Brahney et al., 2014) and even at the global scale (Brahney et al., 2015b). Specifically, Catalan and Camarero (2012) report that increased Saharan atmospheric phosphorus deposition in recent decades has shifted lake phytoplankton from phosphorus limitation to nitrogen limitation in the Pyrenees.

Additionally, under conditions of low nitrogen availability and higher temperatures, N<sub>2</sub>-fixing cyanobacteria would have had a competitive advantage over other algal groups, such as diatoms, leading to an increase in their relative abundance at the expense of other algal taxa. The combined effects of temperature and Saharan dust in our study lake may be amplified for two main reasons. First, Saharan deposition is greater during summer months when aridity is highest. Second, the annual mean temperature rise is mainly driven by spring and summer temperature patterns (Supplementary Fig. S1), leading to a longer ice-free period. This extended exposure to Saharan dust could significantly impact the lake's biota.

### 4.2.3.3 Increase in the relative abundance of cyanobacteria with UV protection mechanisms

Since the 1970s, there has been an increase in the relative abundance of scytonemin, a pigment produced by cyanobacteria that provides effective protection against UV light (Garcia-Pichelt and Castenholz, 1993). Additionally, Chlorophyceae contain zeaxanthin, a protective pigment against excessive radiation (Demmig-Adams and Adams, 2006), which has also increased modestly since the 1970s. Sierra Nevada experiences the highest irradiances and daily doses of photosynthetically

active radiation (PAR), UV-A, UV-B, and biologically effective UV among Spanish mountain ranges (Monforte et al., 2015b, a), due to its high altitude and low latitude (Aphalo et al., 2012). Under this UV-stressful environment, algae typically develop repair mechanisms or photoprotective compounds (Demmig-Adams and Adams, 2006), such as the protective pigments mentioned earlier. Therefore, it is likely that these pigments have increased in response to higher UV radiation intensity. However, in Sierra Nevada, there is evidence that contradicts the expectation of an increasing need for UV protection. Specifically, the reconstructed biologically effective UV-B (280-320 nm) did not show a clear increasing trend from 1913 to 2006 yr CE (Monforte et al., 2015b). Moreover, the period of increased scytonemin coincided with a rise in DOCw (Fig. 7), whose concentration is inversely related to water transparency (Morris et al., 1995), thereby attenuating solar radiation in the water column. Additionally, the Saharan dust intrusions, which intensified during this period, are also known to deliver chromophoric, aromatic, and fluorescent dissolved organic matter (DOM) to European alpine lakes, including those in Sierra Nevada (Mladenov et al., 2009).

It seems reasonable to suggest that the observed increase in scytonemin and zeaxanthin may be linked to the growth of cyanobacteria and Chlorophyceae in radiation-exposed environments within the lake and associated bog (Hauer et al., 1997). These environments include planktonic, epiphytic and/or epilithic habitats and algal mats growing in shallow waters, where protection against UV radiation is crucial. This is supported by current observations of these algae in these areas.

In Borreguil Lake, where light penetrates the entire water column, Chlorophyceae and cyanobacteria may outcompete diatoms due to their higher tolerance to UV light. UV radiation likely has a stronger impact in shallower areas, which experience higher temperatures and greater UV exposure. Additionally, dense mats of benthic chlorophytes may have limited diatom growth through shading and competition for space (DeNicola, 1996). These conditions of elevated temperature and UV exposure may have favoured Chlorophyceae and cyanobacteria over diatoms.

### 5 Conclusions

This study provides novel evidence of ecological changes on a timescale of approximately ~400 years in a high-mountain lake ecosystem within Sierra Nevada National Park, highlighting the sensitivity of algal communities to both climatic variability and atmospheric nutrient inputs. By integrating subfossil pigment analysis with regional climate reconstructions built and calibrated using instrumental data from the Sierra Nevada, our findings underscore the significant role of temperature, precipitation, and phosphorus deposition from Saharan dust in shaping primary producer dynamics.

Our paleolimnological analysis uncovered significant transformations in algal communities, marked fluctuations in chlorophyll-a concentrations, and notable shifts in geochemical proxies spanning the last ~400 years—with the most dramatic changes occurring over the past six decades. During this recent interval, algal biomass reached levels unprecedented in the four-century record, pointing to the intensified influence of contemporary climate dynamics and atmospheric inputs on alpine lake environments. These trends mirrored findings from other paleolimnological investigations at six separate lakes in Sierra Nevada, indicating a synchronic ecological and geochemical response. The data indicate a lake

trend toward prolonged ice-free periods, increased Saharan dust deposition, evapoconcentration, diminished hydrological input, and declining lake levels.

Two major ecological shifts, occurring around ~1840 CE and ~1970 CE, appear to reflect periods of intensified environmental forcing and marked ecosystem responses. The first coincided with the end of the Little Ice Age, signaling an early warming phase and an associated rise in algal productivity. The second, more pronounced shift, reflects the compounded impact of warmer and drier climate conditions and intensified dust-driven nutrient enrichment. This led to a fundamental restructuring of the algal community, with previously unseen abundances of N<sub>2</sub>-fixing cyanobacteria, cryptophytes, and UV-resistant taxa, while diatoms declined.

The algal changes observed in this study are consistent with broader patterns of global climate change. Moreover, they coincide in both timing and direction with transformations reported in other alpine lakes worldwide. In keeping with previous studies, our findings contribute to a better understanding of the climatic and environmental preferences of key algal groups, which can be used for lake classification and as analogues for predicting how algal pigments may respond to future habitat changes. This is particularly relevant in the context of projected increases in temperature and aridity, as well as rising atmospheric nutrient inputs from Saharan dust in regions sensitive to such deposition. The future effects of a warming and drier climate, along with shifts in algal abundance and composition, are likely to have far-reaching ecological consequences. Therefore, this study provides a comprehensive perspective on algal community responses to past environmental changes and may contribute to forecasting future trajectories in similarly vulnerable high-mountain lakes worldwide.

### 690 **CRedIT authorship**

Joana Llodrà-Llabrés: Writing - Original Draft, Conceptualization, Investigation - Data collection, Visualization; Carmen Pérez-Martínez: Writing - Review & Editing, Conceptualization, Investigation - Data collection, formal analysis; Supervision, financial; John P. Smol: Data collection, Writing - Review & Editing; Carsten Meyer-Jacob: Data collection, Writing - Review & Editing; Teresa Vegas: Conceptualization, Financial support, Review and Editing; Javier Sigro: writing - Review & Editing, Conceptualization, Investigation; Teresa Buchaca: Pigment analyses, writing - Review & Editing, Conceptualization, Investigation, Supervision.

### Acknowledgements

This research is part of the project LACEN (OAPN 2403-S/2017) which has been co-funded by the Ministry of Ecological transition in their National Park Autonomous Agency (OAPN) action line. This work was partially funded by grant BIOD22\_001, funded by Consejería de Universidad, Investigación e Innovación and Gobierno de España and Unión Europea – NextGenerationEU. This research is part of the project LifeWatch-2019-10-UGR-01, which has been co-funded by the Ministry of Science and Innovation through the FEDER funds from the Spanish Pluriregional Operational Program

2014-2020 (POPE), LifeWatch-ERIC action line. This work was supported by the **Natural Sciences and Engineering**705 **Research Council of Canada**.

JLL was funded by an Aid for University Teacher Training FPU 2019 (FPU19/04878) by the Spanish Ministry of Universities.

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
