# Peer review of "Tracking striking algal changes over the last ~400 years using subfossil pigments in a high mountain lake (Sierra Nevada, Spain): Have we entered an unprecedented era?"

_EGUsphere, 2025_

## Referee Comment (RC1)

**General comments**

This study presents a new dataset of sedimentary pigments from Lake Borrequil (Sierra Nevada, Spain), in the context of climate change, nutrient sources, and their impact on aquatic productivity in high-mountain lakes. The paper constitutes a valuable contribution to the limited literature on sedimentary pigments as biomarkers of primary producer communities. This paper compares community changes with existing (downscaled) climate data to constrain the environmental factors responsible for the development of the lake ecosystem in the last 440 years.

Overall, these data deserve to be published; however, a clear storyline is difficult to deduce from the large amount of information provided. As we deem this data valuable, we suggest some major changes to enhance the clarity, comparability, attractiveness, and reproducibility of the manuscript.

The paper can benefit from streamlining of the main message e.g., shortening, removal of repetitions and redundant information, restructuring to remove repetitive sections. All of these aspects disturb the flow of the read. Similarly, the number of plots can be reduced; however, there are also plots missing that could aid in understanding the discussion topics and highlight the evidence presented to support the arguments.

Authors often refer to other literature without clearly specifying what they wish to refer to or where the argument is leading. We encourage the authors to pay extra attention to this in the line-by-line comments. Repeating in detail which parts of the literature were relevant would help the reader understand the line of argument, without the need to read all the referenced literature.

**Individual scientific questions/issues ("specific comments")**

The title of this contribution sounds as if it were AI-generated and is overselling. We suggest using an alternative title without AI superficial words, such as "striking", as well as the typical AI-generated format ".:" followed by a question. Here we attach an example of a shorter yet still well-descriptive title suggestion: "Tracking algal changes during the last ~400 years using subfossil pigments in a high-mountain lake, Sierra Nevada, Spain". If authors used LLM/generative AI model(s) extensively, we encourage them to add this in the form of a disclaimer at the end of the manuscript.

Lines 11-14: Ensure that the affiliations are consistent, information such as address, ZIP code, city, state, and country, need to be listed for every institution.

**Abstract**

Please consider shortening the abstract while maintaining focus on the main message within the broader context. Some minor rephrasing can resolve this.

- Line 17 – Suggestion: "have been shown to be" -> "are affected"
- Line 18 – Suggestion: "there is an increasing body of evidence from paleolimnology that indicates" -> "Paleolimnological studies increasingly indicate"
- Line 19 – Suggestion: "paucity" -> "lack"
- Line 19 – This is very specific for it being the second sentence of an abstract to use diatoms. Is there any less specific data in the publication, such as an algal community, or so? It would also better link to the next sentence, addressing the "overall algal community"
- Line 20-22 – Suggestion: "This study investigates the shifts in algal biomass and communities using subfossil sedimentary pigments from the last ~430 years. We use well-dated sediment cores from ..."
- Line 21 – Using high-resolution is problematic, as different methodologies assign different meanings to the term 'high-resolution'. We suggest stating the exact temporal or spatial resolution for clarity.
- Line 23-24 Suggestion: "We noted a significant algal biomass increase in the 1970s, which appears to be linked to increasing temperatures and aerosol concentrations." Thus, the sentence at line 24 becomes repetitive and can be removed.
- Lines 25-26 are somehow repetitive. We suggest merging these sentences into one to avoid redundancy.
- Lines 29-31: As this was already partly mentioned in the sentence at line 24, we recommend adding the enhanced Saharan dust to line 24 and removing this part.
- Line 33 – Did the "N availability" change over time, or is it rather linked to the N/P ratio? If the latter, a clarification is needed here, e.g., "compared to P, which is brought disproportionally to this lake by Saharan dust."
- Line 34-36: We suggest rephrasing to increase the text clarity and remove redundancies, e.g., "The observed changes in the algal community in Borreguil Lake were temporally related to algal community shifts in other Mediterranean sites, suggesting that rising global temperatures, aerosol deposition and enhanced Saharan dust deposition will likely continue to affect the ecological condition of these ecosystems"
- Lines 35-36: It is great to conclude the abstract with the predictions, and we suggest adding the implications or consequences linked to the predictions for the broader context and the importance of studies like this one.

**Introduction**

Overall, the paragraph structure, as well as the structure of the entire section, can benefit from improvements in information distribution.

The novelty of this specific research needs to be more clearly defined, especially in comparison to Jiménez et al. (2018, 2019) and Pérez-Martínez et al. (2020a). The knowledge gap for this study needs

to be better stated by the end of the second paragraph, not the end of the introduction. The studies mentioned above have already answered the first RQ and partially answered the second RQ.

We recommend focusing on the novel aspects of the manuscript and decreasing the number of RQ/hypotheses, which include:

1. the in-depth analysis of community shifts, and

2. The relatively longer timescale perspective.

As is noticeable in the discussion section, the current RQs were also merged and discussed in the two main overall discussion sections. This is a clear sign that there is no need for more than the overall 2 RQs.

We recommend outlining the problem and stating what remains unclear after considering all the other studies addressing aquatic ecosystems under stress in the first and second paragraphs. This text is now placed in paragraphs 5 and 6 instead. Parts of the 3$^{rd}$ and 4$^{th}$ paragraphs can be blended into the "study site" section (2.1).

- Lines 46-47: It is somehow unclear what the stress actually is; a high-altitude location and the position in the Mediterranean zone are not themselves a stress. Could you please clarify what stress you mean already in this sentence?
- Line 51-52: The last sentence of the paragraph brings only information, but no context.
- Lines 53-67 belong to the study site and can be removed here, then blended with the study site section.
- Line 53: A similar phenomenon as in the previous paragraph: the first introductory sentence carries way too little context. Consider adding the crucial argumentation in short already in the introductory sentence, such as here: what makes the lake setting unique? It comes in the paragraph, but it would be great to have it already mentioned in the first sentence.
- Line 58: Is the temperature information from the reference cited in the next sentence? Or where does this come from?
- Line 59: Please provide an estimate of the decline in precipitation, particularly from 1975 to 2020 CE. From what number to what number (in mm) or a percentage.
- Line 64: Where does this information come from? Low-alkalinity – maybe worth mentioning the geological settings of the area.
- Lines 64-65: Repetitive sentence, we suggest removing: "These features make ... and atmospheric factors (refs)".
- Line 68 and elsewhere: Please be careful with the use of "long-term". From a geological perspective, and even in the context of the Quaternary sciences, 430 years is not a long-term period. Although coming from biology, it's understandable why one would use the term 'long-term'. Be aware of the discrepancies in understanding this phrase. And, the sentence on line 68 doesn't bring anything new, thus can be removed.
- Paragraph 5 – This is an important paragraph which would benefit from being mentioned at the beginning of the introduction for the context of the importance of this study.

- Line 70-71: missing references. They occur in the rest of the paragraph, though they should also be listed in this sentence.
- Line 74 and throughout the manuscript: we recommend avoiding vague wording like "linked", "connected", "complex", or "associated"; if something is linked or complex, then please clarify the link or complexity in full.
- Line 77 - "mountain lakes worldwide" - This is somewhat contradictory; the authors state that the Sierra Nevada is unique due to its location. Exposing it to specific Mediterranean drought and warmth, as well as Saharan dust. Other mountain regions are not influenced by those. Yet, this sentence attempts to demonstrate that the shifts in the Sierra Nevada align perfectly with global trends. This would speak against its uniqueness. Please consider resolving this inconsistency in argumentation.
- Lines 81-82: Information in brackets is unusual and should be limited whenever possible in scientific articles.
- Lines 91 - 93: Are these examples relevant to this study?
- Lines 93-94: How is this important in this paragraph? Consider removing this information, as it is already mentioned elsewhere in the manuscript.
- Line 97 – Consider rephrasing: "The atmospheric" replace with "We expect that atmospheric"
- Line 101 – If no other nutrients than P are discussed from this study, consider rephrasing: "and nutrient concentrations – particularly phosphorus" -> "phosphorus concentrations"
- Line 104 – To clearly state what is new in this study, we recommend adding "in more detail (compared to Jiménez xxx)"
- Line 105 – Please be specific and list the "key biogeochemical variables"
- Line 108 - Could you, somewhere in the manuscript, clarify why the period of 430 years was selected? To what extent is this already answered in the previous study by Jimenez xxx? Is there any clear motivation to target this specific time period?
- Lines 107-110: see comments on RQ's, this section would need rephrasing according to the new RQs
- Line 110-112: The last sentence of the introduction section is redundant once the introduction is restructured.

Figure 1:

If possible, we suggest using ArcGIS or QGIS software to display a DEM (SRTM) of Spain and its neighbouring countries instead of relying on Google Maps. Furthermore, four maps are displayed, but only three panels (A, B, and C) are described in the figure caption. The orange box in panel A does not represent the map in panel B. Please provide a scale and north arrow for every map. If Figure 1 aims to show where the lake lies in the larger landscape (region), the right part of panel A is needed. However, if the local landscape around the lake and the lake itself and its catchment and bathymetry are the main goals of this figure, then the figure on the right side of panel A can be removed.

**Methods**

Overall, the methods lack a significant amount of important information and would benefit from additional details for clarity, reproducibility, and repeatability.

**2.1: Study site**

This section could be placed separately, apart from methods, if wanted. As already suggested in the introduction, where the study site is also introduced (that section of the introduction can be shortened, and information can be stated mainly here to eliminate repetitive information), it would be beneficial to introduce the bedrock here, thereby placing the low alkalinity information into context. Now it comes a few sentences later, with no further context.

- Line 119-120: Is this information needed for any of the interpretations? If not, this is redundant.
- Lines 124 – 125: The sentence "This study ... 0.18 ha." deserves to be the first sentence of the section.
- Line 127 – Is there a reference for the watershed vegetation "alpine meadows"? Please add the source of this information.
- Line 129: The physicochemical data are used further in the discussion (e.g., lines 493-498) but are not stated here; only the reference is provided. We recommend clearly stating the data needed for interpretation in this manuscript, so the reader doesn't need to look them up in the cited publication.

**2.2 Sediment coring and sampling**

- Line 136: Is there a particular reason for 26 cm?
- Line 137: What was the motivation for the selected sectioning of 0.25cm and, at a certain depth, switching to 0.5cm? Was there any change in lithology (which is completely absent in this manuscript, see the comments below in the section results)? How, was sectioning directly on-site done? Please add more details on core extrusion.
- Lines 138 – 139: remove "using disposable material to prevent contamination between samples" - this is common sense.
- Lines 139-140: What type of plastic? PET, PP, HDPE? Volume of the vials? Was there an attempt to keep the headspace minimal to avoid pigment degradation by oxygen? How about temperature? Did you store the samples at -20°C whenever possible, or at least at 4°C? All this information is crucial for the method's replicability. These can shortly be added here or stated.
- Line 141: Any reasoning for storing the freeze-dried sample in 4°C instead of a desiccator, frozen, or just at room temperature?
- Lines 140 – 141: remove "for later analyses"

**2.3: Chronology**

Were the data checked for missing inventory? (https://doi.org/10.1016/j.quageo.2020.101106)

- Line 149 – Is the Fukushima accident marker actually used in this study? If not, then this information may be removed.

**2.4: Carotenoid pigment analysis**

- Line 151: Does the number of samples 51 include full chemical replicates (e.g. parallel samples from the core) as well as blanks? Additionally, were the freeze-dried samples stored at room temperature or at the above-mentioned 4°C? It should also be noted that freeze drying has a negative impact on carotenoids; rare carotenoids may not be detected due to degradation during freeze drying. Although these findings have not been published yet, this information should be taken into account when interpreting the pigment data.
- Line 153: Was a cooled centrifuge used? Or how was centrifugation achieved at 4°C?
- Line 154: We encourage authors to specify any modifications made to the HPLC method. Furthermore, the reproducibility of the pigment analysis based on replicates or internal standards is lacking. This would be most necessary, as different carotenoids usually have quite different reproducibility.
- Lines 157-161: Please add references for the interpretations of individual pigments. Perhaps a table with the pigment name, its interpretation, retention time (in the used method), molecular weight, and references would be helpful.
- Line 159: Just out of interest, the references to zooplankton, how do they actually work? I have not seen these pigments particularly associated with zooplankton before. Hence, illustrating the need to outline the interpretational framework here (see later comments in results).
- Lines 161–163: Suggestion: "there is a ... deposited and buried (ref)" -> "carotenoids are less prone to degradation to colourless compounds (ref)."
- Line 164: Were the 10 selected pigments calibrated to concentrations, or only the peak areas used to determine the relative abundance %? Is there any reasoning behind expressing pigments as relative abundances? The closed sum at 100% needs to be justified, as a preselection of pigments is used. If the aim is to compare only specific primary producers, this is possible, but if the aim is to explore shifts of the entire PP community, the absolute concentrations, in umol/DW or umol/OC or OM, would be a better way to present the data. We further encourage authors to report the retention times and relative downcore abundance changes (peak area) of unknown pigments in the supplementary or any open-source database. This helps the pigment community. Additionally, an example of a chromatogram (for example, at 460nm) with annotations may be included in the supplementary data to enhance the reproducibility of the modified HPLC method. This makes the research more reproducible, transparent, usable, and citable.

**2.5: Biological and paleoenvironmental data analysis**

- Line 167-168: Was an existing decalcification method used? If so, please cite it. If not, please specify volumes of added HCl.

- Line 171-172: This sentence could be used as the explanation why these analyses were done, and thus we suggest moving it to the start of this paragraph to make the general statement about C/N.

- Line 172: The reference of Meyers and Ishiwatari (1993) is missing in the reference list and is likely misplaced. Other references are meant here.

- Line 175: Was the isotope analysis done on different samples than C/N? Also, please add the number of samples used/analyzed by both methods, as well as the method of accuracy (error or SD).

- Line 177-179: This sentence is not needed in methods, as the data are discussed later. Please consider our previous comment regarding the use of vague words like "complex".

- Lines 180-183: We encourage authors to introduce both methodologies briefly so that the reader can easily follow the data interpretation. Additionally, it would be advisable to explain the motivation behind using these methods and why relying solely on the TOC from the elemental analysis and sedimentary pigments is insufficient. Are these data, anyhow, comparable to observational data from the current lake water column?

- Line 181 – Please state the full name of the method VRS when it is first mentioned.

- Lines 184-185: The assumption that TOC = DOC is bold, but if it's based on linked monitoring data to surface sediment data, it can be used. Though one should be careful, as with changing vegetation in the watershed, this may not be applicable to older stages of the lake. Additionally, the water TOC model of Meyer-Jacob et al. is based on northern lakes. Is it really applicable to this mountainous lake? A more conservative interpretation should be considered, such as relative changes in TOC to reflect relative changes in DOC in the water. It is more likely that the 1:1 relationship does not exist, as there is at least a fractionation of DOC being recycled, and only a small part is buried: e.g., in the Boreal region, where the carbon burial is expected to be higher, the mean C burial is 22% from the incoming C to the lake: https://doi.org/10.1002/2013JG002345. Ultimately, does the DOC data alter the entire narrative? Isn't the TOC data sufficient?

**2.6 Climate data**

As these builds on previous publications, we suggest keeping this section brief, considering using only one of the two wNAO or SPI to simplify the story and avoid constantly discussing the differences between them. If possible, it would be even better to show the Zr/Al ratios from this nearby lake as a proxy for the deposited dust.

- Line 194: Why was Pearson correlation used? Were the data tested for normality before?

- Lines 209-211: Please clarify the binning method: Were the data binned into arbitrary sampling intervals, and then an average was taken?
- Lines 205 – 208: Please consider rephrasing this long sentence into two shorter ones.

**2.7: Statistics**

The overall impression from this section is that the study includes an excessive number of statistical data treatments without clearly stating the motivation or the need behind using the individual statistical approaches. Every statistical analysis is biased by the selection of variables, and thus, the selection of variables needs to be justified by the aim/RQ that the data should answer. Thus, we strongly recommend using descriptive statistics selectively and with clear objective reasoning.

We encourage authors to make their workflow, as well as data, open source on GitHub or other open-source platforms. This makes the paper stronger, more applicable, citable, and especially reproducible.

- Line 213 - 215 – It may be rephrased as follows: "Before statistical analysis, data were checked for normality and further standardized using a Hellinger transformation to remove asymmetry and incomparabilities in the total variance." We also suggest adding the reasoning behind the selection of the data transformation.
- Line 215: An example of how text can be shortened throughout the manuscript for better readability: "on the basis that" =>"because"
- Line 216-217 – Please state the explanatory variables and add the reasoning behind using the specific data transformation
- Line 219-220 – These sentences can be changed, or placed elsewhere in the paragraph, as we suggest using the context of those further up in the paragraph to clarify the reason why the data were transformed. The d15N sentence can be incorporated further above, e.g. after the sentence about transformation.
- Lines 221-222: The sentence starting "Changes in ... algal groups" can be removed, as it is already introduced in the pigment section
- Lines 222-225: An example of shortening the text for increasing the readability of the section:"The number of significant pigment zones (CONISS; constrained hierarchical clustering; Grimm, 2011) was determined using the broken stick model."
- Lines 227: It is somehow unclear why the data were split, and two separate PCAs were conducted.
- Lines 235: Similar to the comment above, it is not explained why AIC was selected.
- Lines 247-248: "for the period ... CE; 22 samples)" - Yes, this is logical, but it would be better to read at the top before the GLM and RDA details are mentioned. Similar to other parts of this section, the purpose of the used statistics is not well explained.
- Lines 252 – To follow the FAIR principles of publishing scientific results, we encourage authors to publish their code on any open-source platform.

**Results**

To support the readability and clarity of the manuscript's main message, we suggest adding an interpretation in the results section to present the data. In the Results and Discussion section, we keep the line-by-line comments minimal as this section needs restructuring. However, many of the textual comments given above are applicable throughout the manuscript. In a potential second round, we can focus on the Results and Discussion sections in detail.

**For example:**

"N2 fixing cyanobacteria (Aphanizophyll) are sporadically present from x CE to x CE and increase exponentially after x CE. _This rise coincides with low d15N values, both hinting at NOx transformation and fixation._ "

**Another example:**

"Echinenone is rather stable throughout the record, meaning that the production of the pigment is constant. _Either its primary producer (cyanobacteria) or its transformation pathway (degradation of other pigments into echinenone) is constantly operating, regardless of climate and environmental forcing._"

Furthermore, please consider describing the entire dataset (VRS-Chl-a, C/N, dN15, pigments) on a zone-by-zone basis. Alternatively, one can proceed pigment by pigment, as in the examples above, and choose a consistent structure to follow. If the zone-by-zone structure is chosen, a combined interpretation is expected at the end of each zone.

**3.1: Geochronology:**

In a study conducted on a sediment core, the core/sediment picture and description, as well as material characterisation, are expected, yet absent in this manuscript. This may impact several aspects of data interpretation. If possible, we encourage authors to include this information in both the manuscript and the figures, such as the age-depth model.

- Line 259: Please consider displaying both models (CIC and CRS) side by side in the supplementary material. Further, we encourage authors to publish their entire dataset, including the 210Pb and 137Cs data, at any of the open-access data repositories prior to publication, so the data can be linked via DOI to this publication.

**Figure 2:**

We suggest overlaying (A) in (B), or indicating the marker points of 137Cs on the panel B model, as the y-axis is currently not aligned to the same scale. It is advisable to follow the author's guidelines of the journal to fulfil the criteria of readability (e.g., font type and font size). Furthermore, a light grey shading may be used to express the uncertainty intervals. Lastly, as stated in the text, the age-depth model is reliable only until a depth of 6.63cm, but the pigment data are presented all the way to a depth of 14cm.

Thus, the extrapolated ages should be included in the age-depth figure as a dotted line described as "extrapolated".

**3.2 Trends in pigment composition:**

For better readability and message conveying, we suggest presenting the pigment data alongside the chronological description of sediments/lithology and all other proxies. This would affect Figure 3.

- Line 310: It is unclear how the PCA of the climatic data periods explained the total pigment variation. What data were used for this PCA? Was it a merged dataset of pigment data and climate data? If so, this was not clearly explained in the methods.

**Figure 3:**

It is advisable to use colours to distinguish between the different pigments, guiding the reader more easily through the various primary producers. If authors decide to keep the pigments in relative abundance % (which we don't suggest), a stacked bar plot would be easier to constrain the community shifts. Furthermore, plotting the lithology, TOC, C/N, β-carotene, and VRS-inferred Chl-a data along with the pigments would be useful in supporting certain interpretations. Optionally, PC1 & 2 from pigment PCA can also be added, along with the other data, to complete the dataset for interpretation.

For the extrapolated ages, we recommend removing age markers or keeping only 1600 and 1800 as the uncertainty of these ages is high.

It is advisable to follow the journal's guidelines to fulfil the criteria of readability (e.g., font type and font size), as some x-axes have no tick labels. The CONISS clustering dendrogram can be either reduced in width or displayed simply as boxes that delineate clusters.

**Sections 3.3 and 3.4**

We suggest merging sections 3.3 and 3.4 to remove redundancies and clarify the data used for Figure 7. It is unclear whether these are the binned modelled data or if these data originate from the sedimentary record. Further, in the section 3.4, it is recommendable to provide a full description of the SPI, MATA, or refer to another paper? Optionally, these data can be presented in the context of the zone-by-zone description.

**Figure 4:**

Confusing to have two PCA's. If there is no specific reason why these two time periods were split, please consider removing the second PCA; If removing the second PCA, parts of figures 3, 4 and 8 may be merged in one. Furthermore, please consider using a more intuitive temporal scale, such as

whole-round non-diagonal numbers (e.g., 2000, 1950, 1900, 1850, 1800). Similar to other figures, please follow the journal guidelines for figures in terms of fonts, font size, and line width. Additionally, consider using a different line style for PC1 and PC2 to accommodate colour-blind individuals.

Figure 5:

Please refer to the comments above regarding the reduction of PCAs and statistical analysis. Preferably, an objective summary of the full dataset should be presented in the results section.

To enhance the readability of the PCA, we recommend using arrows/vectors for the variables (pigments). Optionally, a colour-coding of the sites/samples by their respective CONISS pigment cluster would help the reader to understand the data and their interpretation. And finally, if the CONISS clusters are used, a confidence ellipse of these clustered data would guide the reader in the statistical significance of the clusters (clearly seen potential overlaps).

Figure 6:

For increased readability, we suggest adding transparency of the individual datapoints and using regular x-label scaling (e.g., 1860, 1870; 1880, 1900). Why are wNAO and SPI showing opposite trends? It is probably best to keep only one metric in the paper; this makes the story shorter and more understandable. Please verify that the SPI unit should be cm [area unit]. month$^{-1}$; as this is data from another publication. Further, we encourage authors to make a general discussion figure, with 3-4 pigments (Diatox, Lut, Echi, Aphanizo), the SPI, MATA, total Chl-a, C/N, N15, timescale boxes marked on it (e.g. LIA, MCA, etc.), and pigment zones marked on it. This would also replace Figures 8 and 9, enhancing the main message delivery to the reader.

Figure 7:

Similar to other figures, please follow the journal guidelines for figures in terms of fonts, font size, and line width. Additionally, an interpretation of the PC axes would be helpful to convey the main interpretation of the data. Since this PCA is performed on the climate data, which is not new in this paper, consider moving this figure to the supplementary material or blending it in the overall discussion plot as line plots.

Section 3.5:

This section could be merged with the chronological description of sediments, pigments, and other proxies. We recommend updating Figure 3 according to our suggestions and referring to it in this text, wherever it will be placed in the restructured results.

Figures 8 & 9:

Please, see the comments in Figure 6.

**Section 3.6:**

Describing PCA and RDA results can be challenging; Good job! Please, try to limit overlap with the zone-by-zone description.

**Figure 10:**

This figure requires revision, as all the names of the pigments are overlapping in the middle of the plot and are not readable. The reasoning behind doing an RDA with only two variables is missing. Please explain the purpose of doing the RDA this way either in the methods or in the results. If forward selection doesn't allow us to use more variables, then we have the option to just show the results of the two drivers in a table. It may also be potentially interesting to perform partial RDA and variance partitioning. [https://pmassicotte.github.io/stats-denmark-2019/07_rda.html#/](https://pmassicotte.github.io/stats-denmark-2019/07_rda.html#/) and show results of variance partitioning (in a table or supplementary).

**Discussion**

In general, the discussion section would benefit from restructuring, grouping and linking processes together to interpret certain phenomena. If the RQ are reduced to 2 as we suggest in the introduction section, the structure of the discussion section must change accordingly. Please consider these changes to improve the clarity and readability of this manuscript. To reduce the number of redundancies, it is preferable to go topic by topic or time zone by time zone; currently, it is a combination of both. The main section headers (4.1 and 4.2) are too long and should be a maximum of 8 words long. Often, a substantial amount of literature is provided, but its relevance and context remain unclear (examples outlined below). It is essential to clearly outline the point one wants to discuss, present the evidence, and highlight the relevant aspects of the literature, ultimately working towards comparison and conclusion. This will enhance the readability of the manuscript

**4.1: increasing biomass?**

**Paragraph 1:**

Lines 419 – 420: Suggestion: "The long-term changes in primary production and algal biomass were assessed using VRS-inferred Chl-a B-car and C/N"

Example of inconsistent paragraph logic structure: This first sentence brings the readers' expectations to the discussion of these biomass and community changes. However, as the paragraph continues, there is no explanation of these changes; instead, it presents a range of values (which belong to the results) and discusses the origin of OM. Thus, we encourage authors to revise the section and paragraph structure in a way that logically groups the information, and the logic is clearly communicated so the reader knows what is connected to what and why.

The interpretation of the C/N ratio follows the common way, but it is advised to consider N-limitation as an alternative explanation. If N is limited, N-fixating cyanobacteria may still result in high productivity and influence the C/N ratio, rather than the changes in the N source. This is also evident in the d15N data. And if pollen data are available from this lake or a nearby lake, these two potential explanations can be distinguished. As Section 4.2.3.2 discusses the N2-fixing cyanobacteria, it would be advisable to link these interpretations together and restructure the discussion to reduce redundancy and repetition. This would also help to streamline the main message and shorten the manuscript.

**Paragraph 2:**

It is essential to clarify the main message that is to be conveyed in this paragraph. Is it: a). Biomass declines in the LIA? or b). Are crustaceans abundant during the LIA? or c). Chl-a and b-car are similar and good indicators of total productivity. The paragraph should focus on a single topic; otherwise, it's very difficult to follow.

Line 433: Could this peak in the pigments be related to better preservation during the cold period? Until here, there is no explanation in the manuscript of how the preservation of the pigments was somehow constrained. As the lake is only 2.8m deep and the light penetrates all the way to the bottom, the preservation of pigments may be a very important factor when interpreting. Maybe the relative changes reflect only the changes in the preservation conditions? If any method or data analysis was used to constrain the preservation, it should be stated in the methods section.

**Paragraph 3:**

This paragraph covers what is after the LIA, but it offers just a description of trends, which is rather results than discussion. There is a lack of context regarding what this means in relation to the RQs. E.g., if biomass follows temperature, then temperature is a driver. When describing a time period, please offer an interpretation. An example could be: *"During the XY timeframe, P was abundant, and xx algae thrived. This appears to be the case across the Sierra Nevada, vouching for a regional driver. Fe/Zr data in study Y shows that the increasing biomass follows dust, here we also find a strong correlation with dust during this time window (table X). We therefore suggest that dust drives the production of biomass."*

**Paragraph 4:**

Paragraph should not be composed of 2 sentences only. Please find an alternative location for this information so it's well-placed in context.

**Paragraph 5:**

This paragraph comprises literature only with no clear structure which would link it to the data. Consider following an order from far away to closer by, or the opposite. Further, this type of information is rather expected in the introduction.

**Paragraph 6:**

Similar to other paragraphs, there is a lack of flow, and it is unclear whether the paragraph addresses the biomass or communities.

Line 457: It would be useful to have the C/N values plotted along with the pigments and the PCs, not only the algal mass. This can be resolved by reorganizing the content of the figures as suggested throughout the review.

Line 460 – Right, now the question is: What does this study add to what is already known?

Line 464 – This part is somehow repetitive and claims that everything is important; if all of it remains vague, what is the conclusion?

**Paragraph 7:**

Line 467 – repetitive information

Line – 469-471: "Longer ice-free … promoting algal growth" - not necessarily, considering that algea bloom in spring, nutrients need to be available in spring. If these nutrients are lying on the ice, and are released at once during spring melt, it only makes algae thrive more. If the dust is not accumulated on the ice, it might sink to the bottom during winter without being used, e.g. a larger part of the load goes to burial.

Line 475 – Please be selective in referencing: e.g., referring to the "United States lakes" – the USA is a big country, and to place this interpretation into a comparable context, it is advisable to specify why and where it is relevant.

Line 478: It is not clear where the data in the supplementary figure S3 originates from. Are these monitoring data? If so, these were not introduced in the sampling section. If they are from another publication, it would still be useful to clarify that those are monitoring data of the water column, not from the sediment.

**Paragraph 8:**

The start of the paragraph is good, but what is missing is the own data part. This should be added to this paragraph in a logical argument.

**4.2 Community changes?**

Line 490-493: The authors introduce here 4 classes of planktonic community, and it is very unclear how these classes were defined, and where these data originate from. Are these from monitoring data as questioned just above? Similarly, for the diatoms, where does the information about periphytic species come from?

Line 493: Why is Ca important? Why not Fe or S? One can argue with any element; thus, it must be justified why Ca is important, if this helps your argument.

Line 500: This may be true, but both diatoms and chrysophytes are observable in the sediment through their siliceous tests. One can then check these under the microscope to confirm the relative changes in their abundances.

Line 508 – These observed algal groups are important, perhaps best mentioned in the methods, and which pigments trace which group in your case.

513-517 - This should have been clear from the methods, and it does not require any additional discussion.

536-537 – If not conclusive, then exclude from the manuscript. This makes it difficult to read.

557-559 – There is a repetitive occurrence of the same conclusion; in general, the whole community seems to be driven by climate change and SPI. In the discussion, one may go beyond this and discuss certain (selective) important exceptions that bring the interpretations closer to the conclusion statements.

564 – This is a rather general statement with little conclusive power.

574-583 – We recommend shortening this literature review to the essential.

Section 4.2.3.3 about UV perception comes way too late. Many aspects of the interpretation can be explained by this phenomenon, but the reader must wait until the very end. Already, the fact that there are more carotenoids than phorbins would hint that the UV is an important driver. Consider restructuring the discussion so that the processes and drivers are more closely linked to the data.

**Conclusions**

Lines 668-671 "During this recent … and geochemical response" may be reduced to "During this recent interval, algal biomass reached levels unprecedented in the four-century record, pointing to the intensified influence of climate warming and dust on Sierra Nevada Lake environments."

Line 671-672 No proxy is mentioned for ice-free periods, Saharan dust deposition in the lake (only from model inferred), evapoconcentration, hydrological input (no XRF), and lake levels. To the best of the readers knowledge, no data has been presented that is deterministic with respect to any of these parameters.

688 "forecasting future projections" – The Authors did not discuss or present how it contributes to future forecasting; this is a somewhat pretentious statement. Just stick to the facts, they are cool enough. There is no need to oversell this research; it is interesting as it is. Quite the opposite, the smaller and more detailed the better. There is none of the Nitrogen and UV stories here, for example.

The data availability section is completely missing at the end of the manuscript. As already mentioned, several times throughout the review, we highly encourage authors to publish their entire dataset at any of the open-access data storage platforms, as well as their statistical and data analysis code on GitHub or any other repository.